# Digital Pregnancy Test Powered by an Air-Breathing Paper-Based Microfluidic Fuel Cell Stack Using Human Urine as Fuel

**DOI:** 10.3390/s22176641

**Published:** 2022-09-02

**Authors:** Irma Lucia Vera-Estrada, Juan Manuel Olivares-Ramírez, Juvenal Rodríguez-Reséndiz, Andrés Dector, Jorge Domingo Mendiola-Santibañez, Diana María Amaya-Cruz, Adrían Sosa-Domínguez, David Ortega-Díaz, Diana Dector, Victor Manuel Ovando-Medina, Iveth Dalila Antonio-Carmona

**Affiliations:** 1Departamento de Energías Renovables, Universidad Tecnológica de San Juan del Río, Av. La Palma No 125 Vista Hermosa, San Juan del Río 76800, Mexico; 2Facultad de Ingeniería, Universidad Autónoma de Querétaro, Santiago de Querétaro 76010, Mexico; 3Departamento de Energías Renovables, Conacyt-Universidad Tecnológica de San Juan del Río, Av. La Palma No 125 Vista Hermosa, San Juan del Río 76800, Mexico; 4Facultad de Ingeniería, Universidad Autónoma de Querétaro, Campus Amealco, Camacho Guzmán, Amealco 76894, Mexico; 5Facultad de Química, Universidad Autónoma de Querétaro, Campus Universitario, Cerro de las Campanas S/N-Edificio 5, Centro Universitario, Querétaro 76010, Mexico; 6Facultad de Ingeniería Química, Universidad Autónoma de San Luis Potosí, Coordinación Académica Región Altiplano (COARA), Matehuala 78700, Mexico; 7Departamento de Botánica, Universidad Autónoma Agraria Antonio Narro, Saltillo 25315, Mexico

**Keywords:** human urine, paper-based microfluidic fuel cell, lateral flow assay tests, pregnancy test, power supply

## Abstract

The direct integration of paper-based microfluidic fuel cells (μFC’s) toward creating autonomous lateral flow assays has attracted attention. Here, we show that an air-breathing paper-based μFC could be used as a power supply in pregnancy tests by oxidizing the human urine used for the diagnosis. We present an air-breathing paper-based μFC connected to a pregnancy test, and for the first time, as far as we know, it is powered by human urine without needing any external electrolyte. It uses TiO${}_{2}$-Ni as anode and Pt/C as cathode; the performance shows a maximum value of voltage and current and power densities of ∼0.96 V, 1.00 mA cm${}^{-2}$, and 0.23 mW cm${}^{-2}$, respectively. Furthermore, we present a simple design of a paper-based μFC’s stack powered with urine that shows a maximum voltage and maximum current and power densities of ∼1.89 V, 2.77 mA cm${}^{-2}$ and 1.38 mW cm${}^{-2}$, respectively, which powers the display of a pregnancy test allowing to see the analysis results.

## 1. Introduction

The use of paper in microfluidic systems has been incorporated for applications such as chemical analysis for their versatility and design at a low cost [1]. Mostly, these microfluidic applications are configured in lateral-flow tests (LFT’s), which allow the capillary flow of an analysis sample (e.g., serum, whole blood, plasma, urine, sweat, saliva, and tissue samples) across a reaction paper strip (rps) without needing an external device, for example, a pump or capillary electrophoresis. An application of LFT’s is as biomedical systems for a quick medical qualitative diagnostic [2,3], such as detection of tuberculosis [4,5,6], infections with streptococcus [7,8], dengue [9], hepatitis B [10], sexually transmitted diseases such as HIV [11,12], and the hormone human chorionic gonadotropin, indicating pregnancy.

However, although the LFT’s have all of the above-mentioned advantages, they present a disadvantage for quantification due to sensitivity and selectivity variation [13]. For this reason, the search for quantitative LFT’s has led to the integration of portable readers, which operate on battery power. An example is the Clearblue Digital Pregnancy Test introduced in 2003, in which the screen displays either “pregnant” or “not pregnant” and indicates how many weeks have passed since conception. Nevertheless, being a single-use LFT, it is discarded with the electronics used for detection, signal processing, signal transmission, a portable reader, and the new battery.

A promising solution to minimize waste pollution and have more portable quantitative devices is to grant autonomy to the LFT’s. The above may be possible if paper-based microfluidic fuel cells (paper-based µFC’s) are used to energize the electronic LFT’s. Some paper-based µFC’s were previously connected to LFT’s as follows:Paper-based µFC’s placed inside the LFT’s using human blood as fuel were reported by Dector, A. et al. [14]Urea and AgNO_3_ paper-based µFC’s placed outside the LFT’s, as was reported by Chino, I. et al. [15]

Paper-based µFC’s are one of the emerging technologies in developing devices with low cost, efficiency, portability, and simple construction for power generation [16]. Further, both cases presented (out- and inside) could grant autonomy for the portable devices when energy is generated with the physiological sample for the diagnosis.

Several works have studied the improvement of paper-based µFC’s by using different anode and cathode catalysts, e.g., noble metal catalysts (Pt/C, Au/C, PtRu/C), enzymes, and bacteria (glucose oxidase, lactate, E-coli), as well as various fuels and oxidants, (e.g., glucose, urea, ethanol, methanol, glycerol) [14,15,16,17,18,19,20,21,22,23], which will be chosen depending on the type of devices or application.

For the development of power sources for pregnancy tests, the physiological fluid used is urine; therefore, the main fuel is the urea in urine. Urea is a good hydrogen carrier; therefore, it could be considered a potential power source with high energy density and stability, is non-toxic, biodegradable, abundant, and has low-cost [24]. Moreover, as urea is found in urine, and the components of urine (over 3000 metabolites) could be used to diagnose numerous diseases [25], the urine could be an excellent alternative as fuel for microfluidic fuel cells. In this sense, the oxidation of urea has been widely studied with several catalysts [26]; for example, Castillo-Martínez et al. [20] evaluated the use of *Escherichia coli* as an anode, which represents an organic catalyst option that showed good results for at least 20 consecutive days. However, using organic catalysts usually presents some disadvantages, such as low performance or needing a complicated immobilization process over the electrodes. The above has led to more extensive research on inorganic electrocatalysts for urea oxidation. Currently, the Ni-catalysts [24,26,27,28,29] are the most common materials developed to achieve an improvement in urea oxidation. Nevertheless, when using urine directly, it is important to use a selective catalyst considering the easy poisoning with all the substances present in urine [30].

In this work, a TiO${}_{2}$-Ni electrocatalyst reported previously by our group [31] was used as an anode due to the synergy created between the TiO_2_ as metallic nano-particle support and the urea oxidation properties of nickel nanoparticles. These TiO${}_{2}$-Ni nanoparticles exhibited excellent behavior as catalysts for urea oxidation, which, as mentioned above, has drawn attention for its availability in the urine. The incorporation of TiO${}_{2}$-Ni as an anode in a paper-based µFC that uses the urine as a urea source could power a quantitative pregnancy test, as J. P. Esquivel predicted when the first paper-based fuel cell was reported [16]. Furthermore, it could be used to develop several portable medical devices.

In this work, we develop an air-breathing paper-based microfluidic fuel cell that incorporates Pt/C as a cathode to reduce the oxygen available in the air and TiO${}_{2}$-Ni as a selective anode for the oxidation of the urea found in the urine. This air-breathing paper-based µFC was collocated inside a pregnancy test to demonstrate that the same physiological sample used for the analysis could also be used to provide power and give autonomy to a portable device. Beyond that, we show the assembly of an air-breathing paper-based µFC’s stack with a series/parallel array that also works with urine as fuel and was connected as a power source of a quantitative pregnancy test. The result is a normal functioning of the pregnancy test, showing the diagnosis on a display powered by an air-breathing paper-based microfluidic fuel cell stack.

The originality of this work lies in the progress toward using direct physiological fluids (human urine) as fuels in air-breathing paper-based microfluidics fuel cells for real applications, leaving behind the ideal evaluation conditions of urea dissolved in an alkaline media. Furthermore, a novel, compact and simple design of an air-breathing paper-based μFC’s stack that provides enough energy to power the display of a quantitative pregnancy test is presented, allowing diagnosing results.

## 2. Materials and Methods

### 2.1. Air-Breathing Paper-Based Microfluidic Fuel Cell Assembly Process

The paper-based µFC was constructed using two electrodes (anode and cathode). These electrodes consisted of pieces of Toray^®^ porous paper electrode (Technoquip Co., Inc., Spring, TX, USA, TGPH-120, 370-µm thick) with 5 mm × 10 mm dimensions. Catalyst ink covered both electrodes, which was prepared by sonicating 73 µL of isopropyl alcohol (J. T. Baker) and 1 mg of catalyst for 15 min. Later, 7 µL of Nafion^®^ 5 % (Sigma Aldrich, San Luis, AZ, USA) were added and mixed by a vortex for another 15 min. Finally, the ink was deposited on the electrode using an airbrush with a final loading of 1 mg cm${}^{-2}$ over half of the surface (5 mm × 5 mm). In this case, the catalyst employed as an anode was TiO${}_{2}$-Ni (synthesized as previously reported [31]); the cathode was of commercial Pt/C (30 wt.% on Vulcan XC-72 from E-TEK).

#### 2.1.1. Air-Breathing Paper-Based µFC’s Placed Inside the Qualitative Pregnancy Test

To evaluate the performance of a simple paper-based microfluidic fuel cell, a pregnancy test (Meditest^®^) was open, and the electrodes were collocated after the reactive pad. The cathode and anode (both with a 5 mm × 5 mm contact area) were placed at the top and bottom, respectively, of the reaction paper strip (rps) to form a “sandwich” arrangement. The cathode was attached on the top to receive oxygen from the environment, and this was possible because the plastic bottom cassette holder of the pregnancy test had a window opening to observe the test line response, which was enlarged and used as an air intake for the cathode, as shown in Figure 1.

#### 2.1.2. Air-Breathing Paper-Based µFC’s Placed Outside the Quantitative Pregnancy Test

A similar process was made with a digital quantitative pregnancy test (Clearblue). In this case, the pregnancy test batteries were removed, and an air-breathing paper-based µFC’s stack was assembled to grant the voltage and current to turn on the display. The paper-based µFC’s stack (Figure 2A) was placed outside the pregnancy test to avoid the modification of the electronics and the system for detecting the hormone, which altogether is relatively complex.

The connection between the fuel cells was a series-parallel arrangement. As shown in Figure 2B, it consists of an array of 2 µFC’s connected in series, then, three of these series arrays are connected in parallel, and all are collocated over one paper strip. The stack is connected directly to the display of the pregnancy test (Figure 2A). The connection in series allows a voltage increment, while the parallel array favors an increase in current. The novelty and simplicity of the stack design rely on the fact that all the electrodes are collocated over the same paper strip; therefore, the fuel is added just once, allowing an easy implementation on LFT’s.

### 2.2. Performance of the Air-Breathing Paper-Based Microfluidic Fuel Cells

The performance evaluation of the paper-based µFC inside the qualitative pregnancy test was based on the oxidation of urea in human urine and the reduction of oxygen in the air. The human urine sample was collected from a young male volunteer using a completely clean and sterile vessel, and the urea concentration of 390.7 mM in the urine was determined by an enzymatic kit (UREA-LQ SPINREACT). This value of urea is in the normal range for a healthy person [30,31,32,33,34].

All electrochemical tests were performed considering the description in the instruction operation manual of the pregnancy test. Further, 60 µL of urine were collocated into the reservoir of the pregnancy test; in this way, the urine was automatically transported through capillary action by the rps and we wet the test line first and then the electrodes (microfluidic fuel cell).

The electrochemical tests were developed as follows: first, a measurement of the OCV (Open Circuit Voltage) as a function of time was made until reaching a stable maximum value; next, a polarization curve (discharge curve) was developed using linear sweep voltammetry at 20 mV s${}^{-1}$ where initial and final potentials were the value of OCV and zero, respectively. Finally, a chronoamperometry was performed to assess the current density stability of the paper-based µFC.

#### Air-Breathing Paper-Based µFC’s Placed Outside the Quantitative Pregnancy Test

The performance evaluation of the stack was evaluated by a discharge curve using linear sweep voltammetry at 20 mV s${}^{-1}$, and the range was established from OCV to zero. The OCV was obtained after collocating the urine in the inlet and getting a stable voltage value (approximately 4 min).

The same sample of the urine of a male human volunteer was used (urea = 390.7 mM) as fuel in the paper-based microfluidic fuel cells stack. Neither the urea solution nor any other solution was used to feed the pregnancy test. As with the qualitative pregnancy test, 60 µL of urine were collocated into the paper-based microfluidic fuel cells, with the urine being transported through capillary action by absorbent paper toward the electrodes, and at the same time, another 60 µL of the same urine were collocated over the pad of the pregnancy test to carry on the biochemical analysis. All tests (performed on qualitative and quantitative pregnancy tests) were measured using a Zahner Zennium potentiostat/galvanostat. The current and power densities were calculated according to the area of the electrodes exposed to the rps. In the first case, the effective area is equivalent to 0.25 cm${}^{2}$, while in the case of the stack, according to the arrangement used, the effective area corresponds to the equivalent of 6 µFC’s (1.5 cm${}^{2}$). The results show an average of 3 evaluation tests using the same human urine sample collected.

## 3. Results and Discussion

### 3.1. Performance Measurement of the Air-Breathing Paper-Based Microfluidic Fuel Cell

The performance of the air-breathing paper-based µFC (configuration: Pt/C and TiO_2_/Ni as cathode and anode, respectively) inside the qualitative pregnancy test (described in the previous section) exhibits the following performance: when the urine is collocated on the rps, the absorption by capillary action is observed in approximately 10 s, then urine makes contact with the conjugate pad and with both electrodes, while the oxygen from the air passes through the porous electrode. Thus, the oxidation–reduction reactions generate the current; the results are presented in Figure 3, and show a typical polarization curve µFC’s. The paper-based µFC shows an open circuit voltage, a maximum current density, and maximum power density of around 0.96 V, 1.00 mA cm${}^{-2}$, and 0.23 mW cm${}^{-2}$, respectively.

The polarization curve, as mentioned above, presents the characteristic behavior of fuel cells. First, a slight decline is observed, related to the reaction kinetics and potential losses due to activation, and a major decline would imply a slower reaction. The second zone refers to the ohmic voltage losses due to the cell’s electronic and ionic resistance; this refers to the connections, materials, and substances in the urine. In Figure 3, the graph clearly shows the relationship between current and voltage in the cell, which is also reflected in the power density curve. Finally, the third zone corresponds to the diffusion zone, related to the transport of fuel and oxidant to the active sites; in this case, as shown in Figure 3, this zone is almost imperceptible, indicating that the transport by capillarity is efficient.

As mentioned above, there is an actual interest in using urine as fuel in microfluidic fuel cells because of their possible application in medical devices. However, for its use, it is important to consider that urine is a complex substance composed of water, creatinine, uric acid, chloride, sodium, potassium, sulfate, ammonium, phosphate, and other ions and molecules in lesser amounts [30]. Therefore, the performance of the µFC used in this work is improved by using a TiO_2_-based catalyst, considering, as is described by Park, S. et al. [35], that the urea oxidation is favored due to the presence of inorganic compounds, such as ${\mathrm{PO}}_{4}^{-3}$. As shown by Dector, D. et al. [31], the synergy of TiO_2_ with Ni also favors the urea oxidation. Urea oxidation has been widely studied in alkaline media according to the following reactions [26]:
$\mathrm{CO}{\left({\mathrm{NH}}_{2}\right)}_{2}+{\begin{array}{c} 6OH \end{array}}^{-}\to {\mathrm{CO}}_{2}+\begin{array}{c} 5H_{2}O \end{array}+N_{2}+{\begin{array}{c} 6e \end{array}}^{-}$$E_{anode}$ = −0.746 V (NHE)$\frac{3}{2}\begin{array}{c} O_{2} \end{array}+\begin{array}{c} 3H_{2}O \end{array}+{\begin{array}{c} 6e \end{array}}^{-}\to {\begin{array}{c} 6OH \end{array}}^{-}$$E_{cathode}$ = 0.4 V (NHE)$\begin{array}{c} \mathrm{CO}{\left({\mathrm{NH}}_{2}\right)}_{2} \end{array}+\frac{3}{2}O_{2}\to \begin{array}{c} {\mathrm{CO}}_{2} \end{array}+\begin{array}{c} 2H_{2}O \end{array}+\begin{array}{c} N_{2} \end{array}$$E_{ceU}$ = 1.146 V (NHE)

However, the presence of all the substances in the urine could promote urea oxidation by different mechanisms; also, some complementary reactions to the urea oxidation could be taking place on the same electrodes. Therefore, the current and voltage vary when human urine is used as fuel compared to that obtained using urea solutions. An advantage of the paper-based µFC presented here is the simplicity of the design and its ease of use, in contrast, for example, to a non-paper-based µFC, which implicates the use of a pumping system. Table 1 shows some of the performance parameters obtained in microfluidic fuel cells (paper and other types) that employ either urea or urine as fuel.

The paper-based µFC’s reported by Chino et al. [15], and by Castillo-Martínez et al. [20], used urea in KOH directly as fuel. The results show high current and power density values (Table 1); however, the conditions of the fuel and electrolytes employed do not resemble human urine; nevertheless, they were important precedents for the use of human urine in paper-based µFC’s. Other microfluidic fuel cells have been used to obtain energy from the oxidation of urea in human urine, focusing principally on the study of catalysts that completely perform urea oxidation. The work of Galindo-de-la-Rosa, J. et al. [32] analyzed the effect of NiAl-layered double hydroxides and PdNiO, while Dector, D. et al. [31] studied modified nanoparticles of TiO${}_{2}$-Ni and used in a photo microfluidic fuel cell. The µFC’s used in these works are more complex than the paper-based µFCs, and need pumps to supply the fuel; therefore, their application in LFT’s does not seem very viable. A paper-based µFC using human urine as fuel was presented by Mankar, C. et al. [33] showing a study of the impact of the electrode’s length. The results obtained in this work with the paper-based µFC collocated inside the pregnancy test and using human urine as fuel present a remarkable improvement in comparison with the results reported by Mankar, C. et al., as seen in Table 1.

As part of the evaluation of the paper-based µFC, a chronoamperometric test using urine as fuel was performed (Figure 4). The test was carried out to observe the current density stability of the paper-based µFC at a potential corresponding to the maximum power density, according to Figure 3 (∼0.45 V). The chronoamperometric curve indicates that the current density obtained decreases over time (5 min) due to the drying of the urine on the reaction paper strip; nevertheless, the time of 5 min (300 s) is enough to produce the electric current density needed in a real application such as pregnancy tests (1 to 3 min to observe results).

The open circuit potential (OCV) (Figure 5) was obtained to evaluate the paper-based µFC capability to recover the OCV after four cycles of charging/discharging. A potential close to the initial value of 0.96 V was reached after each discharging; however, we observed a slight decrease in the voltage (from 0.96 to 0.95 to 0.942 to 0.927 V), probably due to some of the components present in the urine (as suggested with other biological fluids, such as blood) since some of the organic molecules such as creatinine and uric acid could block the electrode surface [35]. Another possible cause could be cathode poisoning, as we discussed in other works because the cathode is on one side exposed to air and on the other side to urine. Therefore, it is exposed to several substances that could block the active sites or even cause a reaction competition on this electrode, which could result in a decrease of performance [36,37]. However, these results (chronoamperometric and charging/discharging OCV) indicate that the device could be employed in a constant short time to power a digital pregnancy test, as will be shown in the next section.

### 3.2. Application-Performance Measurement of the Air-Breathing Paper-Based Microfluidic Fuel Cells Stack

All electrochemical characterizations previously described of the paper-based µFC inside a pregnancy test laid the background for the experimental phase described in this section, which is the direct application of the paper-based microfluidic fuel cell.

An air-breathing paper-based microfluidic fuel cell stack is implemented as the sole power source for a digital pregnancy test (ClearblueR) that normally runs with two 1.5 V alkaline coin batteries, meaning that the µFC’s took the place of the alkaline battery. The same urine sample used before (male, 0.391 M urea concentration) is employed for this proof-of-concept. The digital pregnancy test is connected to the 6-µFC’s series/parallel arrangement, as was described in Section 2.1.2. In this case, as was mentioned, the stack is connected outside the pregnancy test device to not modify the electronics. A urine sample of 60 µL is deposited in the pad of the pregnancy test for the biochemical analysis, and another 60 µL of the same male urine sample (0.391 M urea) is deposited in the paper strip simultaneously to supply the power for the out-reading display.

The performance of the stack used to supply the power to the digital pregnancy test was evaluated with the polarization and power density curves, which are shown in Figure 6. Both curves, polarization, and power density present a similar behavior as described for the µFC inside the pregnancy test, indicating the right functioning of the µFC’s stack. Furthermore, the polarization curve does not show a sudden pronounced fall in the ohmic region, demonstrating no great voltage losses due to the connection between electrodes (stack).

The polarization curve shows that with the presented series-parallel array, the voltage results in approximately 1.89 V, which could be used instead of common alkaline or ion-lithium batteries, while the maximum current and power density are of 2.77 mA cm${}^{-2}$ and 1.38 mW cm${}^{-2}$ (Table 1), respectively.

The power requirements of the out-reader system of the pregnancy test were evaluated with a variable power supply, and the minimum voltage required is 1.9 V and 0.22 mA; therefore, the current (3.18 mA) and voltage (1.89 V) obtained with the stack are enough to power the display, which was also checked experimentally (Figure 2a). The urine sample collocated in the fuel cell stack at the same time as in the pregnancy test provides enough power to turn on the display and show the results of the diagnostic after the ∼3 min that the analysis takes. This shows that the system is stable for enough time to carry out biochemical analysis and is, therefore, a good alternative to grant autonomy in disposable devices besides quantitative pregnancy tests.

A previous work presented by Chino, I. et al. [15] showed an array of three paper-based microfluidic fuel cells connected in series that used urea as fuel and AgNO_3_ as oxidant; the system produced enough voltage to power a pregnancy test screen in place of its battery, however, each one of the µFC’s is made with a different paper strip. In this work, the electrodes are connected in a series/parallel arrangement, and all are collocated over just one paper strip, thus reducing the amount of fuel needed and easing its use and implementation inside an electronic device. The absorption properties of the paper and the transport by capillarity of the urine sample used to feed the µFC’s stack, keep the electrodes wet for enough time to perform the electrochemical reactions, and generate the energy to power the display for almost 4 min. The results show that it is possible to power electronic devices using an air-breathing paper-based microfluidic fuel cell stack and human urine directly as fuel without any other electrolyte. Moreover, it is important to highlight that a simple and compact design of the stack would favor its integration in portable devices.

## 4. Conclusions

This work presents the performance of air-breathing paper-based microfluidic fuel cells (µFC) inside and outside qualitative and quantitative pregnancy test devices using human urine directly as fuel. First, an air-breathing paper-based µFC was collocated inside the pregnancy test, and the human urine sample used for hormone detection was also employed as fuel for the µFC. The power is generated due to the electrochemical reactions on the electrodes: urea oxidation (present in urine) and oxygen reduction (present in air), in the anode and cathode, respectively. To this end, Pt/C was used as the cathode, and a TiO${}_{2}$-Ni catalyst was used as an anode, which has been previously reported as an efficient catalyst to oxidize the urea present in urine. Here, we show that this catalyst oxidizes the urea even in the presence of all the substances that have urine. The performance parameters also improve over the paper-based microfluidic fuel cell fed with human urine. Besides, the fuel cell design presented here was easily collocated inside the pregnancy test, which would allow a future integration in LFT’s.

The µFC’s stack was constructed based on the performance of the single air-breathing paper-based µFC and was connected (outside) to a digital pregnancy test. The stack consists of six µFC’s connected in a series/parallel array and was fed with human urine without any other electrolyte. The stack presented an open circuit voltage, a maximum current density, and a maximum power density of ∼1.89 V, 2.77mA cm${}^{-2}$, and 1.38 mW cm${}^{-2}$, respectively. According to the power requirements of the display, the current generated by the µFC’s stack is almost an order of magnitude higher. Therefore, the display could work without problems for as long as the diagnosis finishes; in this case, that means almost for 3–4 min. Another advantage of the proposed air-breathing paper-based µFC’s stack is its simple and compact design. In this stack, all the electrodes are connected in one paper strip, and since we are dealing with single-flow microfluidic fuel cells, its operation depends on just collocating the physiological fluid (urine) in one inlet without any other electrolyte; and for its implementation inside the device, would be enough by having a window for the air intake considering that the electrodes (cathodes) are porous, so the oxygen is taken directly from the air and there is no need for any external electrolyte to its transport.

The work presented here shows that air-breathing paper-based microfluidic fuel cells could easily work using human urine directly and, therefore, are an appropriate power source for biomedical lateral flow assays, such as pregnancy tests, which in conjunction with the future development of electronic systems completely made from paper, and the advance of more efficient biochemical techniques for quantitative diagnoses could lead to more affordable and global access to healthcare. Furthermore, this work considers reducing the environmental impact of disposing batteries that were used just once.

## Figures and Tables

**Figure 1 sensors-22-06641-f001:**
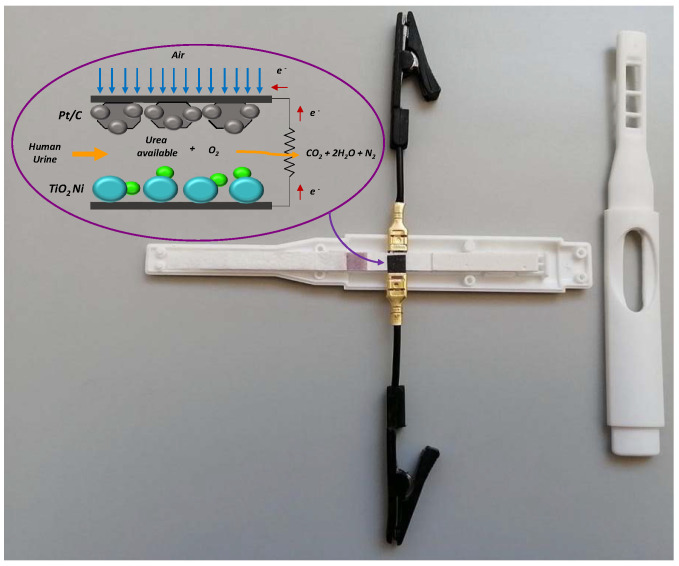
Photograph and schematic design of the qualitative pregnancy test used as a support for an air-breathing paper-based microfluidic fuel cell.

**Figure 2 sensors-22-06641-f002:**
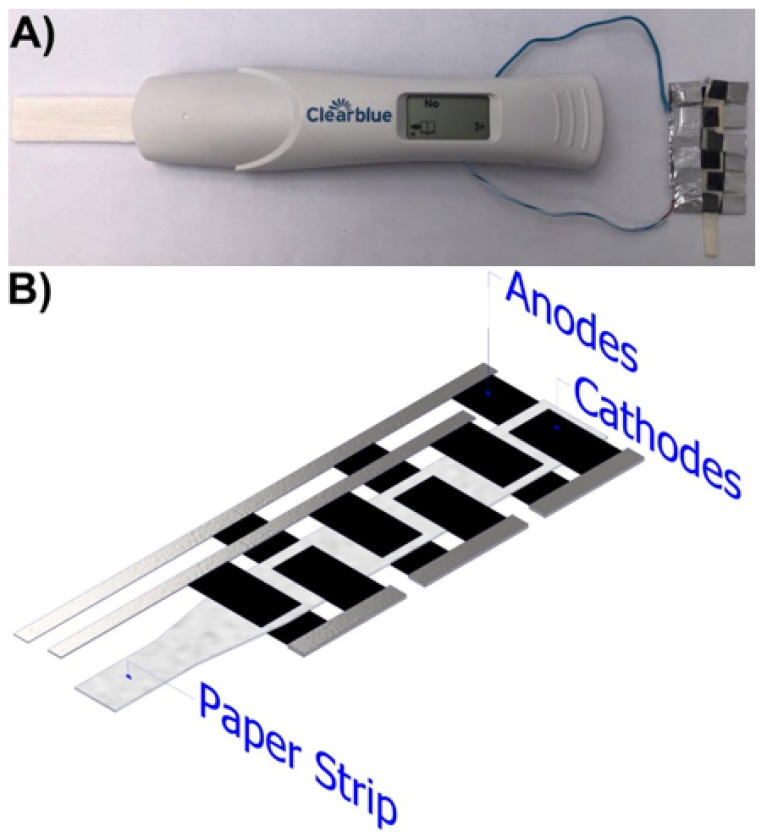
(**A**) Air-breathing paper-based microfluidic fuel cell stack (6-cell series/parallel arrangement) using urine as fuel and air as oxidant placed outside the quantitative pregnancy test. (**B**) Scheme of the series/parallel arrangement of the paper-based microfluidic fuel cell stack.

**Figure 3 sensors-22-06641-f003:**
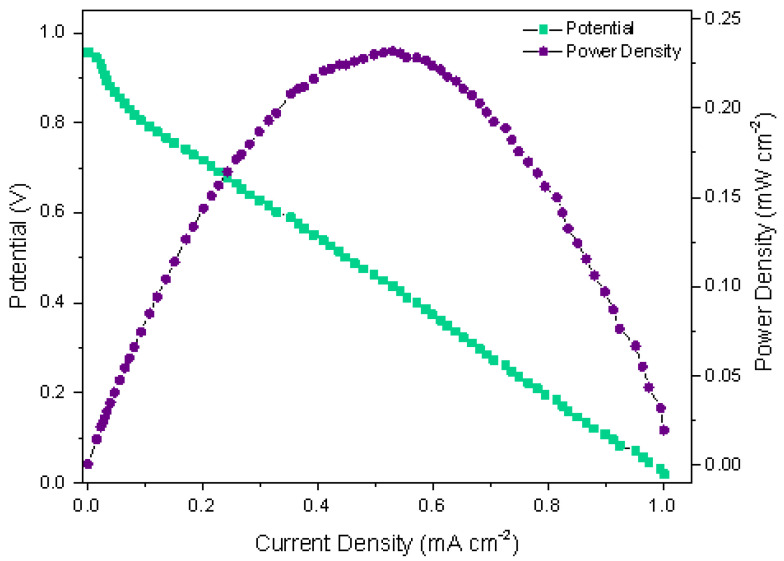
Polarization and power density curves of the air-breathing paper-based µFC with TiO_2_/Ni anode and Pt/C cathode inside the qualitative pregnancy test using human urine as fuel.

**Figure 4 sensors-22-06641-f004:**
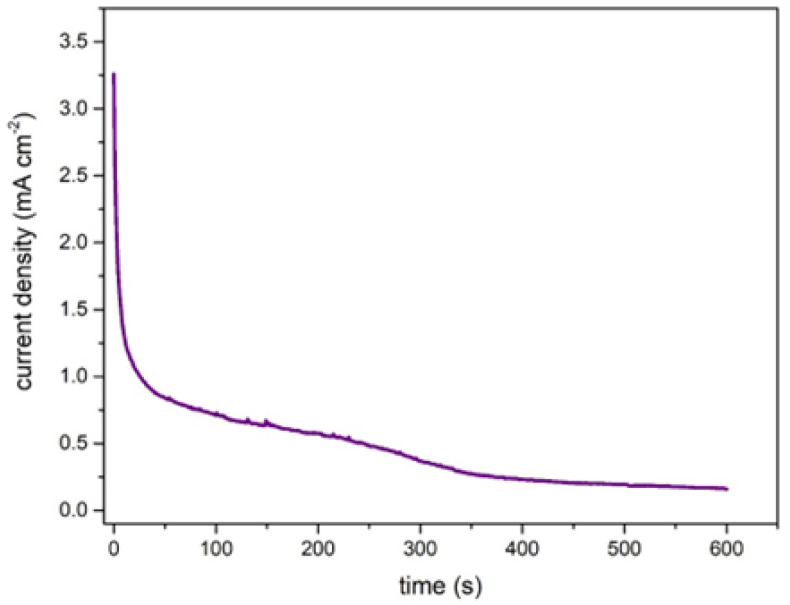
Chronoamperometry of the air-breathing paper-based µFC with TiO_2_/Ni anode and Pt/C cathode inside the qualitative pregnancy test using urine as fuel.

**Figure 5 sensors-22-06641-f005:**
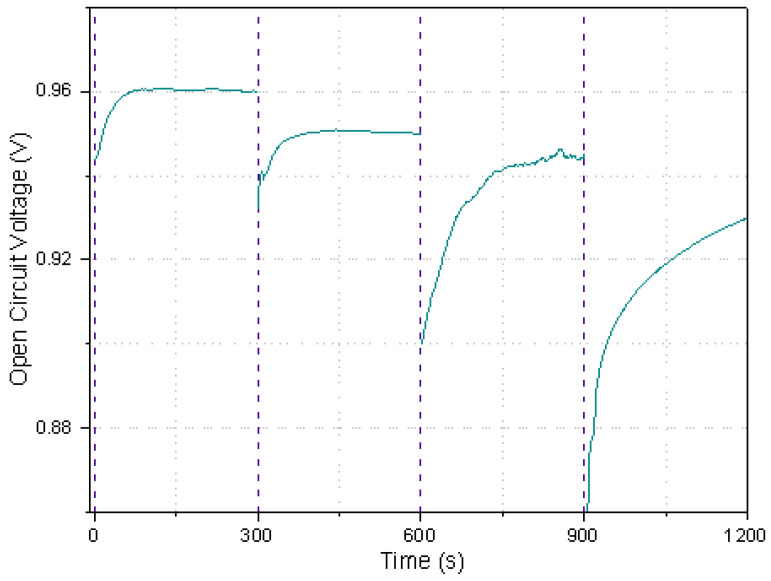
Open circuit voltage stability, using urine as fuel for an air-breathing paper-based µFC with TiO_2_/Ni anode and Pt/C cathode inside the qualitative pregnancy test.

**Figure 6 sensors-22-06641-f006:**
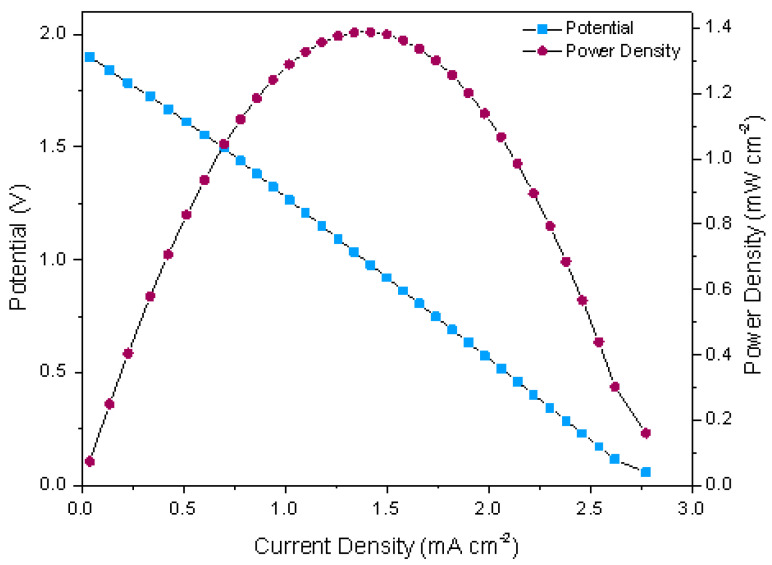
Polarization and power density curves of the air-breathing paper-based µFC stack with TiO_2_/Ni anode and Pt/C cathode outside the digital pregnancy test using human urine as fuel.

**Table 1 sensors-22-06641-t001:** Performance parameters of urea or urine microfluidic fuel cells reported in the literature.

Electrodes (Anode-Cathode)	Fuel	µFCs Type	OCV (V)	$J_{max}$ (mA cm${}^{-2}$)	$W_{max}$ (mW cm${}^{-2}$)	Reference
Pt-C	Urea (0.3 M) in KOH (1 M)	Paper µFCs	0.6	4.3	0.91	[15]
Pt/C-*E. coli*	Urea (0.33 M) in KOH (0.3 M)	Paper-based bacterial µFCs	0.83	3.253	0.608	[20]
Ag-C	Human urine (pH = 6) in H_2_O_2_ (9.78 M)	Paper-based microfluidic microbial FCs	0.5	0.56	0.1288	[33]
NiAl–LDHs–Pt/C	Human urine (0.7973 M urea)	air breathing PMMA µFCs using pump	1	122	50	[32]
TiO${}_{2}$-Ni–Pt/C	Human urine (0.366 M urea)	Photo PMMA µFCs using pump	0.7	1.7	0.09	[31]
TiO${}_{2}$-Ni–Pt/C	Human urine (0.391 M urea)	Air-breathing paper-based µFCs inside pregnancy test	0.96	1	0.23	This work
TiO${}_{2}$-Ni–Pt/C	Human urine (0.391 M urea)	Air-breathing paper-based µFCs stack outside pregnancy test	1.89	2.77	1.38	This work
**LDHs:** Layered Double Hydroxides						
**PMMA:** Polymethylmethacrylate						

## Data Availability

Not applicable.

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
