# Peer review of "Digital Pregnancy Test Powered by an Air-Breathing Paper-Based Microfluidic Fuel Cell Stack Using Human Urine as Fuel"

_sensors, 2022, doi:10.3390/s22176641_

Round 1

Reviewer 1 Report

Dear Authors,

Many thanks for your work. I have the following comments:

- in your manuscript, you have mentioned that you are presenting a paper-based μFC integrated in a pregnancy test for the first time, however, you already cited an article that states the first paper [15] [https://doi.org/10.1016/j.jpowsour.2018.06.082 ]

- you claim that the originality of this work lies in progress towards the use of microfluidics fuel cells in real applications, ex. to provide enough energy to turn on the display on a pregnancy test, allowing to see the test results. This has been done also by chino et. al. in the same article [15]. honestly, I don't see any novelty in your work, the proof of concept has been demonstrated in their article.

Author Response

Thanks for your comments, we gladly present an aswer to them in the attach document.

Reviewer 2 Report

Estrada et al. reported a paper fuel cell for powering pregnancy test based on the red-ox reaction of urea in human urine samples. They improved the performance of the paper fuel cell using TiO2 – Ni as the anode and powered a digital pregnancy test. I recommend the publication of this work after addressing the following comments.

1.       How does this paper fuel cell tolerate different concentrations of urea in human urine? In this study, the urine was from one donor. For real world applications, will the difference in urea concentration affect the performance of the fuel cell?

2.       Can the electrode be regenerated after testing? How does the cost of the paper fuel cell compare to the Li ion battery?

3.       What’s the readout the clearblue digital quantitative pregnancy test? Is the result a certain value or a word description of positive or negative?

4.       Is there any difference in performance for the clearblue test result between Li battery power and urine power? Any comparison experiments?

5.       The manuscript needs extensive editing for the language and typos.

Author Response

Thanks for your comments, we gladly present an answer to them in the attach document

Reviewer 3 Report

The authors integrate a paper-based microfluidic fuel cell with a pregnancy lateral flow test. Paper-based microfluidic fuel cells are versatile in terms of anode/cathode catalysts, fuels, and oxidants. The authors used a published microfluidic fuel cell system that uses urea in the urine as the fuel source to power a pregnancy test. Specifically, the authors used human urine instead of urea dissolved in an alkaline media. Also, a novel and simple design of a microfluidic fuel cell stack that could provide energy sufficient to power the pregnancy test was established.

This work is interesting. It is, however, not clear if the authors will use the power generated by the fuel cell for quantitative analysis or just qualitative analysis. Please clarify. It does not make much sense to use the system if only a qualitative evaluation of pregnancy is conducted. Nevertheless, the authors should emphasize the importance of using a quantitative pregnancy test which requires significant external power in comparison with a qualitative pregnancy test. In addition, quite a few English errors were found in the manuscript; some of which are confusing and may cause trouble in understanding the context. For example, lines 34-35. These issues have to be addressed before being considered for publication.

Round 2

Reviewer 1 Report

No more comments 

Reviewer 2 Report

Publish as is.

Reviewer 3 Report

The context remains confusing because of poor written English. English editing by a native English user is strongly recommended.